# FEATURE SELECTION WITH NEURAL ESTIMATION OF MUTUAL INFORMATION

## ABSTRACT

We describe a novel approach to supervised feature selection based on neural estimation of mutual information between features and targets. Our feature selection filter evaluates subsets of features as an ensemble, instead of considering one feature at a time as most feature selection filters do. This allows us to capture sophisticated relationships between features and targets, and to take such sophisticated relationships into account when selecting relevant features. We give examples of such relationships, and we demonstrate that in this way we are capable of performing an exact selection, whereas other existing methods fail to do so.

## 1 INTRODUCTION

We describe a novel approach to supervised feature selection based on neural estimation of mutual information between features and targets. We call our method *MINERVA*, Mutual Information Neural Estimation Regularized Vetting Algorithm.

Feature selection methods are distinguished in two classes: wrappers, and filters. On the one hand, wrappers assume knowledge of the learning model and use the learning process as a subroutine. They are usually computationally expensive, and they are model-dependent. On the other hand, filters utilize a score of dependence between features and target, and they select a subset of features based on this score. Filters do not require knowledge of the learning procedure, and hence they are model-independent.

MINERVA belongs to the class of filters, and utilizes the mutual information as score.

Estimating the mutual information between random variables is challenging. The classical estimator is the Kraskov-Stögbauer-Grassberger (KSG) estimator introduced in Kraskov et al. (2004), and proved to be a consistent estimator in Gao et al. (2018). Recently, a modern, consistent estimator called *Mutual Information Neural Estimator (MINE)* was proposed in Belghazi et al. (2018), and applications have flourished. Our feature selection procedure MINERVA utilises MINE to compute the mutual information score. Our score evaluates subsets of features as an ensemble, instead of considering one feature at a time as most feature selection filters do. This allows us to capture sophisticated relationships between features and targets, and to take such sophisticated relationships into account when selecting relevant features. We give examples of such relationships, and we demonstrate that MINERVA is capable of performing an exact selection, whereas other existing methods fail to do so.

The paper is organised as follows. Section 2 recalls the fundmentals of MINE, the Mutual Information Neural Estimator. Section 3 explains our method of featurte selection and describe our neural network architecture. Section 4 presents our numerical experiments with MINERVA. Finally, Section A collects the proofs of the lemmata and propositions of the article.

## 2 NEURAL ESTIMATION OF MUTUAL INFORMATION

In this section, we recall the neural estimation of mutual information following Belghazi et al. (2018).

Let $\mathcal{X} \subset \mathbb{R}^d$ and $\mathcal{Y} \subset \mathbb{R}^e$ represent sample spaces. Let $X$ and $Y$ be random variables taking values in $\mathcal{X}$ and $\mathcal{Y}$ respectively. Let $P_{XY}$ denote the joint distribution of $X$ and $Y$; and let $P_X \otimes P_Y$ denote the product of the marginal laws of $X$ and $Y$.

The mutual information $I(X;Y)$ of $X$ and $Y$ is defined as the Kullback-Leibler divergence between $P_{XY}$ and $P_X \otimes P_Y$.

Using the Donsker-Varadhan representation of the Kullback-Leibler divergence (see Donsker & Varadhan (1983)), we can write

$$I(X;Y) = \sup_{f} \quad E_{P_{XY}}[f(X,Y)] - \log\left(E_{P_X \otimes P_Y}[\exp(f(X,Y))]\right), \tag{1}$$

where $E_{P_{XY}}$ denotes expectation with respect to $P_{XY}$, $E_{P_X \otimes P_Y}$ denotes expectation with respect to $P_X \otimes P_Y$, and the supremum is taken over all measurable functions $f : \mathcal{X} \times \mathcal{Y} \to \mathbb{R}$ such that the two expectations are finite.

Given samples

$$(x_1, y_1), \ldots, (x_n, y_n), \tag{2}$$

from the joint distribution of $X$ and $Y$, we can use the representation in equation (1) to estimate the mutual information of the two random variables. Indeed, we can represent the functions $f$ in equation (1) via a neural network $f_\theta$ parametrised by $\theta \in \Theta$, and then run gradient ascend in the parameter space $\Theta$ to maximise the emprirical objective functional of equation (1), where the first expectation is replaced by

$$\frac{1}{n}\sum_{i=1}^{n} f(x_i, y_i),$$

and the second expectation is replaced by

$$\frac{1}{n}\sum_{i=1}^{n} \exp\left(f(x_i, y_{\sigma(i)})\right),$$

where $\sigma$ is a permutation used to shuffle the $Y$-samples and hence turn the samples of equation (2) into samples from $P_X \otimes P_Y$.

The described approach to estimate the mutual information $I(X;Y)$ is at the core of MINE, and we will rely on this method to construct a feature selection filter.

## 3 FEATURE SELECTION METHOD

In this section, we describe our method of feature selection.

Let $\mathcal{X} \subset \mathbb{R}^d$ and $\mathcal{Y} \subset \mathbb{R}^e$ represent sample spaces. Let $X$ and $Y$ be random variables taking values in $\mathcal{X}$ and $\mathcal{Y}$ respectively. We interpret $Y$ as the target of a prediction / classification task, and we interpret $X$ as a vector of features to use in this prediction / classification task.

Given $n$ samples

$$(x_1, y_1), \ldots, (x_n, y_n),$$

from the joint distribution $P_{X,Y}$, a permutation $\sigma \in S_n$, a real valued function $f : \mathcal{X} \times \mathcal{Y} \to \mathbb{R}$, and a $d$-dimensional vector $p \in \mathbb{R}^d$, we write

$$\mu(f,p) = \frac{1}{n}\sum_{i=1}^{n} f(p \odot x_i, y_i),$$

$$\nu(f,p) = \frac{1}{n}\sum_{i=1}^{n} \exp\left(f(p \odot x_{\sigma(i)}, y_i)\right), \tag{3}$$

where $p \odot x_i$ is the Hadamard product of $p$ and $x_i$. We use $\mu(f,p)$ to approximate $E_{P_{XY}}[f(p \odot X, Y)]$, and we use $\nu(f,p)$ to approximate $E_{P_X \otimes P_Y}[\exp(f(p \odot X, Y))]$,

Let $f_\theta$, $\theta \in \Theta$ be a familily of measurable functions

$$f_\theta : \mathcal{X} \times \mathcal{Y} \to \mathbb{R}$$

parametrised by the parameter $\theta \in \Theta$ of a neural network. Let $p \in \mathbb{R}^d$ be a $d$-dimensional vector. We define

$$v(\theta, p) = -\mu(f_\theta, p) + \log(\nu(f_\theta, p)) \tag{4}$$

and, recalling Section 2, we consider $v(\cdot, p)$ as an approximation of the negative of the mutual information of $p \odot X$ and $Y$. Moreover, for non-negative real coefficients $c_1, c_2, a$ we define

$$\ell(\theta, p, c_1, c_2, a) = v(\theta, p) + c_1 \left\| \frac{p}{\|p\|_2} \right\|_1 + c_2 (\|p\|_2 - a)^2, \tag{5}$$

where $\|\cdot\|_1$ denotes $L^1$-norm and $\|\cdot\|_2$ denotes $L^2$-norm.

The function $\ell$ is the loss function. It consists of three terms. The first term $v(\theta, p)$ is the discretisation of the functional that appears in the Donsker-Varadhan representation of the Kullback-Leibler divergence. It approximates the negative mutual information between the target and the $p$-weighted features.

The second term $\left\| \frac{p}{\|p\|_2} \right\|_1$ is a regularisation term on the weights $p \in \mathbb{R}^d$. It induces sparsity by pushing to zero the weights of non-relevant features.

Finally, the third term $(\|p\|_2 - a)^2$ controls the euclidean norm of the weights $p \in \mathbb{R}^d$ by penalising the square of the difference between said norm and the target norm $a$. This is meant to prevent the weights of relevant features from diverging.

Our feature selection method consists in finding a minimiser $\hat{\theta}$ of

$$\theta \longmapsto v(\theta, \mathbf{1}),$$

where $\mathbf{1} = (1, \ldots, 1) \in \mathbb{R}^d$, and then using this $\hat{\theta}$ as the initialisation of the gradient descent for the minimisation of

$$\theta, p \longmapsto \ell\left(\theta, p, c_1, c_2, \sqrt{d}\right).$$

We stop this gradient descent when the estimated mutual information between the weighted features and the targets becomes smaller than the mutual information that corresponds to the minimiser $\hat{\theta}$. After the gradient descent has stopped, we select the features that correspond to non-null weights, i.e. to non-null entries of $p$. More precisely, our method is described in Algorithm 1.

The architecture of the neural network used in the parametrisation of the test functions $f_\theta$ is represented in Figure 1.

We implement our MINE-based feature selection in the [pypi]-package *minerva*.

---

**Algorithm 1** Mutual Information Neural Estimation Regularized Vetting Algorithm

---

**Require:** random variables $X \in \mathcal{X}$, $Y \in \mathcal{Y}$, hyperparameters $r > 0$, $c_1 \geq 0$, $c_2 \geq 0$.
1: $\theta \leftarrow$ initialise network parameters
2: **repeat**
3:     Draw $n$ samples $(x_1, y_1), \ldots, (x_n, y_n)$ from the joint distribution $P_{XY}$
4:     Sample shuffling permutation $\sigma$ from $S_n$
5:     Update $\theta \leftarrow \theta - r \nabla_\theta v(\theta, \mathbf{1})$
6: **until** convergence
7: Initialise $\varphi \leftarrow \theta$, $p \leftarrow \mathbf{1}$.
8: **repeat**
9:     Draw $n$ samples $(x_1, y_1), \ldots, (x_n, y_n)$ from the joint distribution $P_{XY}$
10:     Sample shuffling permutation $\sigma$ from $S_n$
11:     Update $\varphi \leftarrow \varphi - r \nabla_\varphi \ell(\varphi, p, c_1, c_2, \sqrt{d})$
12:     Update $p \leftarrow p - r \nabla_p \ell(\varphi, p, c_1, c_2, \sqrt{d})$
13: **until** convergence
14: **return** $\{i : |p_i| > 0\}$

---

Figure 1: Neural network architecture

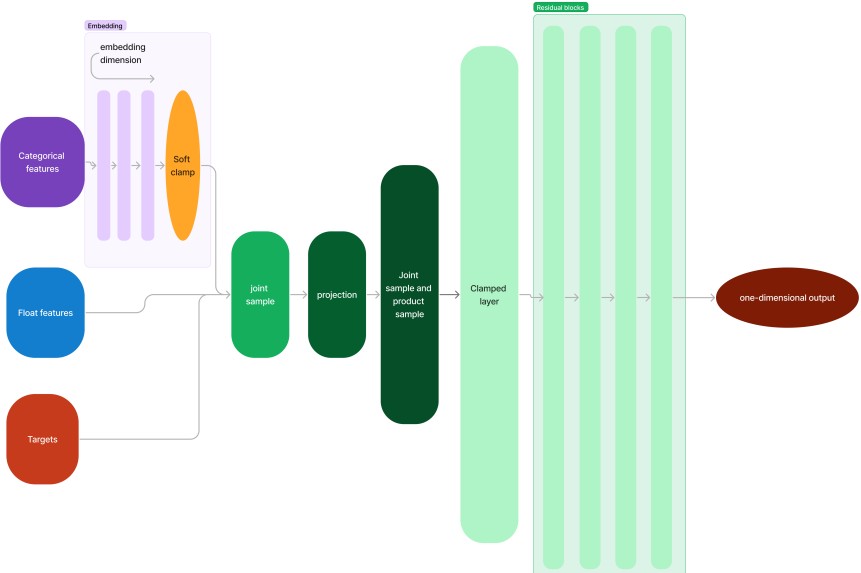

## 4  EXPERIMENTS

In this section, we present the results of our numerical experiments. Our experiments are based on synthetic data.

With synthetic data, out of the $d$-features to select from, we know the subset $t \subset \{1, \ldots, d\}$ that the target depends on, and thus we can evaluate feature selection methods by reconciling their selection with $t$. More precisely, let $s$ be a subset of $\{1, \ldots, d\}$. We say that the selection of the features $s$ is exact if $s = t$, and it is non-exact otherwise. If the selection is non-exact, either $t \not\subset s$ or $s \supsetneq t$. In the former case, we say that the non-exact selection is of type I; in the latter case, we say that the non-exact selection is of type II. Non-exact selections of type I compromise the downstream prediction task because they subtract information relevant for the prediction. Non-exact selections of type II might not reduce the dimensionality of the problem, but they do not compromise downstream tasks.

### 4.1  OUR SYNTHETISED $X$-$Y$ RELATIONSHIP

We study the phenomenon whereby a target $Y$ depends on whether two independent discrete random variables $X_{k_0}$ and $X_{k_1}$ are equal or not.

On the one hand, this sort of dependence is relevant in practice. Assume, for example, that you are dealing with a data set recording international money transfers. This data set will have one column $X_{k_0}$ recording the currency of the country from which the transfer is sent, and another column $X_{k_1}$ recording the currency of the country to which the transfer is sent. The distribution of your data will depend on whether the transfer is multicurrency or not, namely on whether $X_{k_0} = X_{k_1}$.

On the other hand, this sort of dependence is not well captured by existing feature selection filters. We demonstrate this in a first small example. Then, we assess the performance of our feature selection filter on a synthetic dataset that is meant to be representative of a common regression learning tasks, and that embeds the dependence that existing feature selection filters cannot capture.

| method | selected | expected | evaluation |
|---|---|---|---|
| mutual_info_classif | $1, \ldots, 30$ | $3, 8$ | non-exact type II |
| pyHSICLasso | $1, \ldots, 30$ | $3, 8$ | non-exact type II |
| minerva | $3, 8$ | $3, 8$ | exact |

Table 1: Experiment 1.A - Comparison of three different feature selection methods.

### 4.1.1 EXPERIMENT A

Let $d$ be a positive integer, and let $m > 2$ be a positive integer larger than 2. For $i = 1, \ldots, d$ let $X_i$ be a random positive integer smaller that or equal to $m$. The random variables $X_1, \ldots, X_d$ are assumed independent and identically distributed. Fix two integers $1 \leq k_0 < k_1 \leq d$ and define

$$Y = \mathbb{1}\{X_{k_0} = X_{k_1}\} = \begin{cases} 1 & \text{if } X_{k_0} = X_{k_1} \\ 0 & \text{otherwise.} \end{cases} \tag{6}$$

We consider the task of predicting $Y$ from the vector $(X_1, \ldots, X_d)$, and we want to select from this vector the features that are relevant for the prediction.

In this context, feature selection methods that rely on a metric $h(X_i, Y)$ of dependence between each feature $X_i$ and the target $Y$ are bound to fail. This is explained in Lemma 1: the pair-wise assessment of $(X_i, Y)$ cannot possibly produce an exact selection because $Y$ is independent from every $X_i$. It is only by considering the ensemble of features $X_1, \ldots, X_d$ that we can produce an exact selection.

We confirm that in our numerical experiments. We test our MINE-based feature selection method against two benchmarks.

The first benchmark is sklearn.feature_selection.mutual_info_classif.[1] This method estimates the mutual information $I(X_i; Y)$ for all $i = 1, \ldots, d$ and it selects those features $k$ such that $I(X_k; Y) > \epsilon$ for a given threhsold $\epsilon \geq 0$. The estimation of $I(X_i; Y)$ is based on the KSG estimator, introduced in Kraskov et al. (2004).

The second benchmark is HSIC Lasso, see Yamada et al. (2014). This method selects features $i_1, \ldots, i_k$ that correspond to non-null entries of the maximisers of

$$\beta \mapsto \sum_{i=1}^{d} \beta_i h(X_i, Y) - \frac{1}{2} \sum_{i,j=1}^{d} \beta_i \beta_j h(X_i, X_j),$$

where $h$ is the Hilbert-Schmidt independence criterion introduced in Gretton et al. (2005). We use the implementation of HSIC Lasso given in pyHSICLasso.[2]

Table 1 summarises our findings. As expected, neither sklearn.feature_selection.mutual_info_classif nor pyHSICLasso were able to complete an exact selection. Their selections are non-exact of type II. Notice moreover that if we force pyHSICLasso to select two features only, the selection is non-exact of type I. Instead, MINERVA was able to complete an exact selection.

**Lemma 1.** *Let $m > 2$ be a positive integer. Let $X_1, \ldots, X_d$ be independent identically distributed with $P(X_1 = n) = 1/m$ for $n = 1, \ldots, m$. Let $k_0$ and $k_1$ be two distinct positive integers smaller than or equal to $d$, and let $Y$ be as in equation* (6). *Then, for all $i = 1, \ldots, d$*

$$I(X_i; Y) = 0, \tag{7}$$

*namely $X_i$ and $Y$ are independent. Moreover,*

$$I(X_{k_0}, X_{k_1}; Y) = \frac{m-1}{m} \log\left(\frac{m}{m-1}\right) + \frac{1}{m} \log m, \tag{8}$$

*and $I(X_{k_0}; Y|X_{k_1}) = I(X_{k_1}; Y|X_{k_0}) = I(X_{k_0}, X_{k_1}; Y)$.*

---

[1] See https://scikit-learn.org/stable/modules/generated/sklearn.feature_selection.mutual_info_classif.html

[2] See https://pypi.org/project/pyHSICLasso/

| method | selected | expected | evaluation |
|---|---|---|---|
| mutual_info_regression | 14, 18, 19, 20, 23, 25, 28, 31, 34, 38 | 6, 8, 14, 18, 19, 20, 23, 24, 28, 31 | non-exact type I |
| pyHSICLasso | 4, 11, 14, 18, 19, 20, 23, 24, 28, 31 | 6, 8, 14, 18, 19, 20, 23, 24, 28, 31 | non-exact type I |
| Boruta | 14, 18, 19, 20, 23, 24, 28, 31 | 6, 8, 14, 18, 19, 20, 23, 24, 28, 31 | non-exact type I |
| minerva | 6, 8, 14, 18, 19, 20, 23, 24, 28, 31 | 6, 8, 14, 18, 19, 20, 23, 24, 28, 31 | exact |

Table 2: Experiment 1.B - Comparison of four different feature selection methods.

| method | number of features | in-sample R2 | out-of–sample R2 |
|---|---|---|---|
| no selection (use all features) | 40 | 0.8615 | 0.7990 |
| mutual_info_regression | 10 | 0.7647 | 0.6980 |
| pyHSICLasso | 10 | 0.7717 | 0.7004 |
| Boruta | 8 | 0.7669 | 0.7023 |
| minerva | 10 | 0.8799 | 0.8469 |

Table 3: Experiment 1.B - Accuracy of a gradient boosting model trained on the features selected by various methods

#### 4.1.2 EXPERIMENT B

Let $d_1, d_2$ be positive integers. Let $X_1, \ldots, X_{d_1}$ be i.i.d random variables such that $P(X_1 = k) = 1/m$ for $k = 1, \ldots, m$, for some positive integer $m > 1$. Let $X_{d_1+1}, \ldots, X_{d_1+d_2}$ be i.i.d random variables with uniform distribution on the unit interval. It is assumed that $X_1, \ldots, X_{d_1}$ and $X_{d_1+1}, \ldots, X_{d_1+d_2}$ are independent. Let $k_0, k_1$ be distinct positive integers smaller than or equal to $m$. Let $n < d_2$ and let $d_1 < j_0 < \cdots < j_n \leq d_1 + d_2$ and $d_1 < i_0 < \cdots < i_n \leq d_1 + d_2$. We define

$$Y = \begin{cases} \sum_{\ell=1}^{\ell=n} \alpha_\ell \sin \left(2\pi X_{j_\ell}\right) & \text{if } X_{k_0} = X_{k_1} \\ \sum_{\ell=1}^{\ell=n} \beta_\ell \cos \left(2\pi X_{i_\ell}\right) & \text{otherwise.} \end{cases} \tag{9}$$

In other words, $Y$ is a non-linear function of some of continuous features if $X_{k_0} = X_{k_1}$, and $Y$ is some other non-linear function of some other conitnuous features if $X_{k_0} \neq X_{k_1}$.

We consider the task of predicting $Y$ from the vector $(X_1, \ldots, X_{d_1}, X_{d_1+1}, \ldots, X_{d_1+d_2})$, and we want to select from this vector the features that are relevant for the prediction. This setup is a combination of a straightforward feature selection setup where a target depends non-linearly on a subset of features, and the sort of dependence utilised in Experiment A. Namely, we assume that there are two continuous non-linear functions $f_1$ and $f_2$, and the target is a transformation through $f_1$ of some of the continuous features if two discrete variables happen to be equal, and it is a transformation through $f_2$ of some other continuous features if those two discrete variables are not equal.

We test MINERVA against two benchmark filters, and one bechmark wrapper.

The first benchmark filter is sklearn.feature_selection.mutual_info_regression.[3] The second benchmark filter is HSIC Lasso as implemented in pyHSICLasso.[4] The benchmark wrapper is Boruta, as implmented in arfs.[5]

Table 2 summarises our findings. MINERVA is the only method capable of completing an exact selection. Sklearn's mutual information, HSIC Lasso, and Boruta perform a non-exact selection of type I. This is reflected in the prediction of the target given the selected features: out-of-sample accuracy of a gradient boosting model trained on MINERVA's selection decisively outperfoms the accuracies of the same model trained on the features selected by the other methods. See Table 3.

---

[3]See https://scikit-learn.org/stable/modules/generated/sklearn.feature_selection.mutual_info_regression.html
[4]See https://pypi.org/project/pyHSICLasso/
[5]See https://pypi.org/project/arfs/

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

## A  PROOFS

*Proof of Lemma 1.* For ease of notation, take $k_0 = 1$, $k_1 = 2$. We only need to prove equation (7) for $i = k_0, k_1$. For integers $i, y$, let

$$a(y, i) = \mathbb{1}(y = i) = \begin{cases} 1 & \text{if } y = i \\ 0 & \text{otherwise.} \end{cases}$$

For $x_1 = 1, \dots, m$ and $y = 0, 1$ we have

$$P(Y = y | X_1 = x_1) = \begin{cases} P(X_2 \neq x_1) & \text{if } y = 0 \\ P(X_2 = x_1) & \text{if } y = 1 \end{cases} = \frac{m-1}{m} a(y, 0) + \frac{1}{m} a(y, 1)$$

Therefore,

$$\begin{aligned}
P(Y = y) &= \sum_{x_1=1}^{m} P(X_1 = x_1, Y = y) \\
&= \sum_{x_1=1}^{m} P(Y = y | X_1 = x_1) P(X_1 = x_1) \\
&= \frac{1}{m} \sum_{x_1=1}^{m} \left( \frac{m-1}{m} a(y, 0) + \frac{1}{m} a(y, 1) \right) \\
&= P(Y = y | X_1 = x_1)
\end{aligned}$$

where on the last line $x_1$ is any positive integer smaller than or equal to $m$. We conclude that

$$\begin{aligned}
I(X_1; Y) &= \sum_{x_1=1}^{m} \sum_{y=0}^{1} P(X_1 = x_1, Y = y) \log \left( \frac{P(X_1 = x_1, Y = y)}{P(X_1 = x_1) P(Y = y)} \right) \\
&= \sum_{x_1=1}^{m} \sum_{y=0}^{1} P(X_1 = x_1, Y = y) \log \left( \underbrace{\frac{P(X_1 = x_1, Y = y)}{P(X_1 = x_1) P(Y = y | X_1 = x_1)}}_{=1} \right) \\
&= 0.
\end{aligned}$$

The equality $I(X_2; Y) = 0$ is proved in the same way.

Finally, we establish equation (8). For integers $x_1, x_2$, let $b(x_1, x_2) = 1$ if $x_1 = x_2$, and $b(x_1, x_2) = 0$ otherwise. Then, for positive integers $x_1, x_2 \leq m$ and $y = 0, 1$, we can write

$$P(Y = y | X_1 = x_1, X_2 = x_2) = a(y, 0)(1 - b(x_1, x_2)) + a(y, 1)b(x_1, x_2),$$

and

$$
\begin{aligned}
P(X_1 = x_1, X_2 = x_2, Y = y) &= P(Y = y | X_1 = x_1, X_2 = x_2)P(X_1 = x_1, X_2 = x_2) \\
&= \frac{1}{m^2}\Big(a(y, 0)(1 - b(x_1, x_2)) + a(y, 1)b(x_1, x_2)\Big),
\end{aligned}
$$

and

$$P(X_1 = x_1, X_2 = x_2)P(Y = y) = \frac{1}{m^2}\left(\frac{m-1}{m}a(y, 0) + \frac{1}{m}a(y, 1)\right).$$

Let $c(x_1, x_2, y) = a(y, 0)(1 - b(x_1, x_2)) + a(y, 1)b(x_1, x_2)$. Plugging these in the definition of the mutual information between $(X_1, X_2)$ and $Y$, we conclude

$$
\begin{aligned}
I(X_1, X_2; Y) &= \frac{1}{m^2}\sum_{x_1,x_2=1}^{m}\sum_{y=0}^{1}(c(x_1, x_2, y))\log\left(\frac{c(x_1, x_2, y)}{\frac{m-1}{m}a(y, 0) + \frac{1}{m}a(y, 1)}\right) \\
&= \frac{1}{m^2}\sum_{x_1,x_2=1}^{m}\left((1 - b(x_1, x_2))\log\left(\frac{m(1 - b(x_1, x_2))}{m-1}\right) + b(x_1, x_2)\log\left(mb(x_1, x_2)\right)\right) \\
&= \frac{1}{m^2}\sum_{x_1=1}^{m}\left((m-1)\log\left(\frac{m}{m-1}\right) + \log(m)\right) \\
&= \frac{m-1}{m}\log\left(\frac{m}{m-1}\right) + \frac{1}{m}\log m.
\end{aligned}
$$

$\square$