# OpenReview forum: "Feature selection with neural estimation of mutual information"
_ICLR.cc/2024/Conference — Submitted to ICLR 2024_

### Official Review · Reviewer_5aJX · 2023-10-29

**Soundness:** 1 poor
**Presentation:** 1 poor
**Contribution:** 1 poor
**Rating:** 1
**Confidence:** 5

**Summary:**

This papers proposes a novel feature selection algorithm based on estimating the mutual information by using neural networks. The authors propose to test it against a synthetic dataset, showing that the algorithm is able to successfully select the important features.

**Strengths:**

Feature selection experiments with synthetic data should always be used to estimate the performance of the algorithms.

**Weaknesses:**

- The paper writing needs to be improved. Some sentences appear not to be finished. There is a lack of coherence in the explanation.
- The introduction section omits one of the feature selection classes: the embedded methods. No information regarding these methods is provided.
- Multiple mathematical terms are not correctly defined:
    - $p$ is defined after it was used.
    - $S$ is never defined.
    - The target norm $a$ in Eq. 5 is replaced by $\sqrt{d}$ in the algorithm, but $d$ is never defined.
- No information regarding the training procedure was provided.
- The loss function never takes into account the correlation between $X$ and $Y$. Thus, I assume the trained network will provide a constant output that minimizes the loss function, which make it totally independent on the data provided to the algorithm.
- The experimental section does not consider neither recent filter methods (like Inf-FS [1] or ILFS [2], por instance) nor other feature selection methods like CAE [3] or E2E-FS [4].
- There is no experimental results over real datasets, despite the fact that the paper is only 6 pages (3 extra pages could be used for providing a better algorithm explanation as well as an extensive experimental section).


[1] Roffo, G., Melzi, S., & Cristani, M. (2015). Infinite feature selection. In Proceedings of the IEEE international conference on computer vision (pp. 4202-4210).

[2] Roffo, G., Melzi, S., Castellani, U., & Vinciarelli, A. (2017). Infinite latent feature selection: A probabilistic latent graph-based ranking approach. In Proceedings of the IEEE international conference on computer vision (pp. 1398-1406).

[3] Balın, M. F., Abid, A., & Zou, J. (2019, May). Concrete autoencoders: Differentiable feature selection and reconstruction. In International conference on machine learning (pp. 444-453). PMLR.

[4] Cancela, B., Bolón-Canedo, V., & Alonso-Betanzos, A. (2022). E2E-FS: An End-to-End Feature Selection Method for Neural Networks. IEEE Transactions on Pattern Analysis and Machine Intelligence.

**Questions:**

Can you provide an explanation about the missing parameters and their meaning?

---

### Official Review · Reviewer_Mxn5 · 2023-10-30

**Soundness:** 2 fair
**Presentation:** 2 fair
**Contribution:** 2 fair
**Rating:** 3
**Confidence:** 4

**Summary:**

A new supervised feature selection method, utilizing neural mutual information estimation, is presented in this paper. This method considers feature subsets as a whole. It achieves precise feature selection, outperforming competing methods in two synthetic datasets.

**Strengths:**

The authors introduce a method that selects features based on neural estimation of mutual information.

The method outperforms existing techniques in two synthetic datasets.

**Weaknesses:**

The experiments were limited to two synthetic datasets, casting doubt on the method's applicability to real-world datasets. Multiple factors can influence model performance. These include sample size, hyperparameters $c_1$ and $c_2$, the neural network architecture, and the noise that cannot be explained by the input $\mathbf{X}$. Without a comprehensive analysis of these factors, the general efficacy of the proposed method remains uncertain.

Equation (1) provides an estimate for mutual information. Yet, once the constraints are introduced in Equation (5), it's unclear if the function still acts as a mutual information estimator. This requires more in-depth theoretical exploration.

The definition of $\phi$ in Line 11 of Algorithm 1 is unclear.

**Questions:**

1. Does the proposed method work in real-world datasets?

2. Is Equation(5) an estimator of mutual information?

3. How if $\phi$ in Line 11 of Algorithm 1 defined?

---

### Official Review · Reviewer_R5qG · 2023-10-30

**Soundness:** 2 fair
**Presentation:** 2 fair
**Contribution:** 2 fair
**Rating:** 3
**Confidence:** 4

**Summary:**

Authors of this paper propose mutual information neural estimation regularized vetting algorithm (MINERVA) for supervised feature selection based on neural estimation of mutual information between features and targets. The feature weights are introduced into mutual information criterion with sparse regularizations. Experiments are performed on synthetic data.

**Strengths:**

Mutual information is used as a learning criterion for supervised feature selection.

**Weaknesses:**

The proposed model is built on existing work by introducing feature weights in the neural estimation of mutual information, while the proposed sparsity regularization needs to be properly explained.

Experiments are conducted only on two synthetic datasets, and experimental details are missing. Hence, the experimental results are difficult to be justified.

**Questions:**

The regularization on p in (5) contains two terms: l1_norm on l2-normalized p and the square loss between l2-norm of p and a constant a. It looks like the proposed l1_norm term is on the scaled-invariant p, which leads to unbounded p. So, the square loss term is introduced to prevent diverging. It is unclear what is the benefits of introducing the l1_norm on scaled p comparing to l1_norm on p. Moreover, it is unclear how the constant a should be set for different datasets.

Authors mentioned that the gradient descent stops when the estimated mutual information becomes smaller than non-selection model. In Algorithm 1, it mentioned the termination happens when convergence. It is unclear which gradient descent method is used. From Algorithm 1, it seems that some sort of stochastic gradient descent is applied. If so, it may not be reasonable to use the mentioned stop criterion since the selection step may not proceed if one stochastic gradient step cannot make any progress to minimize the initialized objective.

The conventional gradient descent method seldom set some entries of p to zero values unless some projection is done after descent step. To select feature using non-null entries of p or in step 14 of Algorithm 1 might not be practical.

The metric h(X_i, Y) is not defined in page 5.

The experimental settings can be described in detail. For example, some important statistics of the data are missing, such as the number of samples are sampled, the dimension of the synthetic data, how the synthetic data are generated, and the reported results on how many random runs of the experiments.

One key drawback is that only synthetic datasets are used, so it is unclear how the proposed model behaves on real data. For real data, it is often unknown which features are exact, so evaluation of feature selection is often dependent of the downstream prediction tasks. Although Table 3 demonstrate some results, it is hard to judge due to missing experimental details for example, why different methods are compared with different number of features and  what is the in-sample and out-sample ratio.

---

### Meta-Review · Area_Chair_KPUZ · 2023-12-05

**Metareview:**

All reviewers remrked that this paper has several severe problems, both on the conceptual side and on the experiments side. There is no rebuttal, so I recommend rejection of this paper.

**Justification For Why Not Higher Score:**

There seem to be substantial problems with this paper, and without any rebuttal, it is not clear how these problems could be overcome.

**Justification For Why Not Lower Score:**

N/A

---

### Decision · Program_Chairs · 2024-01-16

Reject